# How age influences phonotaxis in virgin female Jamaican field crickets (*Gryllus assimilis*)

Karen Pacheco, Jeff W. Dawson, Michael Jutting and
Susan M. Bertram

Department of Biology, Carleton University, Ottawa, Ontario, Canada

## ABSTRACT

Female mating preference can be a dominant force shaping the evolution of sexual signals. However, females rarely have consistent mating preferences throughout their lives. Preference flexibility results from complex interactions of predation risk, social and sexual experience, and age. Because residual reproductive value should theoretically decline with age, older females should not be as choosy as younger females. We explored how age influences phonotaxis towards a standard mate attraction signal using a spherical treadmill (trackball) and a no-choice experimental protocol. Female Jamaican field crickets, *Gryllus assimilis*, were highly variable in their phonotaxis; age explained up to 64% of this variation. Females 10 days post imaginal eclosion and older oriented toward the mate attraction signal, with 10- and 13-day females exhibiting the greatest movement in the direction of the signal. Our study suggests 10- and 13-day old females would be most responsive when quantifying the preference landscape for *G. assimilis* sexual signals.

## INTRODUCTION

Mate choice is a dominant force shaping the evolution of sexual signals (*Andersson, 1994*). Substantial research has focused on identifying which signals are preferred and how preferences maintain signal variation (*Jacot, Scheuber & Brinkhof, 2007*; *Jennions & Petrie, 1997*; *Ritchie, 1996*; *Verburgt, Ferguson & Weber, 2008*). Historically, many researchers assumed individuals make consistent mating preferences throughout their lives, resulting in relatively invariant selection influencing sexual signals (*Andersson, 1994*). However, individuals often exhibit flexible mating preferences (*Kodric-Brown & Nicoletto, 2001*; *Moore & Moore, 2001*; *Lynch et al., 2005*). Preference flexibility may result from complex interactions of many intrinsic and extrinsic factors including predation risk (*Magnhagen, 1991*; *Godin, 1995*), competition for available mates (*Fawcett & Johnstone, 2003*), parasite load (*Lopez, 1999*), social and sexual experience (*Dugatkin, 1992*; *Milinski & Bakker, 1992*; *White & Galef, 2000*; *Collins, 1995*; *Marler, Foran & Ryan, 1997*; *Kasumovic, Hall & Brooks, 2012*), genetic makeup (*Bakker & Pomiankowski, 1995*), and age (*Juventin, LeQuette &*

Corresponding author
Susan M. Bertram,
Sue.Bertram@carleton.ca

*Dobson, 1999*; *Qvarnstrom, Part & Sheldon, 2000*; *Kodric-Brown & Nicoletto, 2001*; *Moore & Moore, 2001*; *Coleman, Patricelli & Borgia, 2004*; *Lynch et al., 2005*; *Lynch et al., 2006*).

Theoretically, residual reproductive value should decline with age (*Roff, 1992*; *Stearns, 1992*). Older females (with low residual reproductive value) may not be as choosy as younger females (with high residual reproductive value), because the benefits gained from being choosy may be offset by the increased costs of maintaining the preference (*Real, 1990*; *Parker, 1983*; *Milinski & Bakker, 1992*; *Jennions & Petrie, 1997*). Costs associated with being choosy include time and energy in sampling, predation risk, and territorial quality (*Jennions & Petrie, 1997*). *Moore & Moore (2001)* demonstrated this in cockroaches (*Nauphoeta cinerea*). They found that older females were less choosy than younger females and the cost of being choosy (taking longer to select a mate) increased rapidly beyond an optimal age. Females past this optimal age required less courtship to mate and produced fewer offspring (*Moore & Moore, 2001*).

Our study focuses on how age influences female field cricket phonotaxis. Male crickets defend territories from rivals and use these territories to broadcast acoustic mate attraction signals (*Alexander, 1962*). Males signal acoustically by rubbing their forewings together. Each closing stroke produces a pulse of sound; multiple pulses are concatenated into chirps (*Bennet-Clark, 2003*). Females exhibit phonotaxis towards male acoustic mate attraction signals and select between potential mates based on variation in male body size, mate attraction signals, courtship signals, aggressive behavior, and contact pheromones (*Leonard & Hedrick, 2009*; *Rebar, Bailey & Zuk, 2009*; *Thomas & Simmons, 2010*; *Bailey et al., 2011*; *Beckers & Wagner, 2011*; *Verburgt, Ferreira & Ferguson, 2011*; *Deb, Bhattacharya & Balakrishnan, 2012*; *Hedrick & Kortet, 2012*; *Stoffer & Walker, 2012*).

*Prosser (1994)* was the first to investigate how age influenced female cricket mating preferences. *Prosser, Murray & Cade (1997)* revealed that older females (*Gryllus texensis*; aged 25–28 days post imaginal eclosion) exhibited greater movement toward a signal than younger females (11–14 days post imaginal eclosion) in one-speaker trials. Interestingly, both younger and older females showed strong preferences in three-speaker trials. Given older females responded to both trial types, older females seemed more motivated to mate. Conversely, younger females appeared more selective, only exhibiting preferences in trials with multiple mate opportunities (*Prosser, Murray & Cade, 1997*). *Prosser, Murray & Cade*'s (*1997*) study suggests age influences mating preferences and highlights the importance of exploring preference across a broad range of ages.

Most age studies have incorporated mating experience into their experimental design (e.g., *Kodric-Brown & Nicoletto, 2001*; *Mautz & Sakaluk, 2008*). While it makes sense to combine age with mating experience because the two are often positively correlated (*Forslund & Part, 1995*; *Prosser, Murray & Cade, 1997*; *Moore & Moore, 2001*; *Martin & Hosken, 2002*; *Mack, Priest & Promislow, 2003*), teasing apart the effect of age becomes more difficult. *Judge, Tran & Gwynne (2010)* used a fully factorial experimental design to explore the relative importance of age and mating experience in female *G. pennsylvanicus* and revealed that mating experience was more important than age in influencing mating

preferences. They suggest that because females are likely to mate quickly in the wild, any virgin age effects should be short-lived in nature.

Here we quantify how age influences female phonotaxis in the Jamaican field cricket, *G. assimilis*. Despite *Judge, Tran & Gwynne*'s (*2010*) findings on the importance of mating experience, we chose to study only virgins because we needed to determine the best age to quantify virgin female phonotaxis for a subsequent study. We assessed phonotaxis slightly beyond the natural range of ages within which females are likely to be reproductively active in the wild (1–28 days post imaginal molt; (*Zuk, 1987*; *Cade, 1979*)). We quantified phonotaxis because females acoustically orient toward male mate attraction signals prior to mating (*Alexander, 1962*; *Loher & Dambach, 1989*). We used a no-choice protocol and monitored phonotaxis toward a standard male acoustic attraction signal. Each female was tested only once so that our results were not confounded by experience or habituation. We used a spherical treadmill (trackball) to quantify phonotaxis (Kramer spherical treadmill or locomotion compensator: *Kramer, 1976*; *Weber, Thorson & Huber, 1981*; *Doherty, 1985*; *Doherty & Pires, 1987*, trackball: *Hedwig & Poulet, 2005*; *Hedwig, 2006*. When mounted on the spherical treadmill, the cricket could walk in any direction with little or no encumbrance. We quantified each female's locomotion over a set period of time with respect to a standard mate attraction signal that was broadcast through a nearby speaker. Based on life history theory, we predicted older virgin crickets would show greater movement toward the standard signal than younger females. Our findings provide a relative standard for the degree of virgin female *G. assimilis* responsiveness at a variety of ages. They also reveal the most responsive age range to use when quantifying the preference landscape for *G. assimilis* sexual signals.

## METHODS

Adult *G. assimilis* were collected at the Stengl "Lost Pines" Biological Station (University of Texas at Austin), in Bastrop County, Texas, United States (N 30°5′ 14.046″, W 97°10′ 28.988″) from 15–24 September 2008 and transferred to the greenhouse laboratories at Carleton University, Ottawa, Ontario, Canada. *Gryllus assimilis* were placed in communal plastic bins (64 × 40 × 42 cm) with a 16:8 h L:D illumination period at 28 ± 2°C and fed *ad libitum* water [provided in individual plastic containers (11 cm diameter × 3.5 cm height) filled with marbles to provide a surface for the crickets to stand on] and food (Harlan's Teklad Rodent diet 8604M; 24.3% protein, 40.2% carbohydrate, 4.7% lipid, 16.4% fiber, 7.4% ash; Harlan Laboratories, Madison, WI, USA). Individuals mated freely and females were provided with moist sand [provided in individual plastic containers (11 cm diameter × 3.5 cm height)] in which to lay their eggs. Juveniles developed in separate communal bins to minimize the likelihood of larger crickets cannibalizing smaller crickets.

In 2012 we separated, 200 juvenile female crickets from the communal cricket bins shortly after their ovipositor became visible. We housed these juveniles together and monitored them daily for imaginal eclosion. Newly eclosed adult females were housed individually in 520 mL (11 cm diameter × 7 cm height) clear plastic containers with

screened lids (4 × 4 cm mesh screens). We provided each female with a small piece of cardboard egg container for shelter and *ad libitum* water and food (as described above); light cycles and temperatures were identical to the communal rearing environment. We randomly assigned each female to an age group: 1-, 4-, 7-, 10-, 13-, 16-, 19-, 22-, 25- or 28-day post imaginal eclosion (hereafter day). We chose to use a large number of age categories so we could quantify the age(s) at which the females were exhibited the highest phonotaxis for a subsequent study on mating preferences. Females were tested for their phonotaxis to a standard mate attraction signal on the day they reached the age of their assigned age group.

## Standard mate attraction signal

We created a standard mate attraction signal (hereafter standard signal) using Adobe Audition CS5.5 software. We fashioned this standard signal after results published in *Whattam & Bertram (2011)*. This study showed that male acoustic mate attraction signaling is affected by diet; as such we selected parameters that reflect the mean signal parameters from a population of males reared on high quality food (with the exception that *Whattam & Bertram (2011)* recorded their songs at 26°C, while we played these songs at 22–23°C). The standard signal's parameters were: carrier frequency = 3719 Hz, pulse duration = 10.14 ms, interpulse duration = 15.21 ms, pulses per chirp = 8, interchirp duration = 1055 ms; broadcast at an intensity of 60.6 dB SPL (re: 20 μPa RMS) as measured using an EXTECH Digital Sound Level Meter (Model # 407768) with probe pointing directly at the active speaker, directly above the spherical treadmill, 17 cm from the active speaker (Fig. 1).

## Spherical treadmill construction and calibration

The inspiration for creating the spherical treadmill for quantifying cricket locomotor behavior came from the earlier development of the "Kramer Treadmill" (*Kramer, 1976*) and its extensive use by Huber and colleagues in their seminal studies of the neural basis of cricket phonotaxis (*Weber, Thorson & Huber, 1981*; *Thorson, Weber & Huber, 1982*; *Huber et al., 1984*). This approach has seen various degrees of refinement and re-invention as technologies have improved and studies have evolved with different needs (e.g., *Mason, Oshinsky & Hoy, 2001* for tracking tachinid fly orientation with respect to cricket calling song stimuli). The principal idea of a Kramer Treadmill is that it allows the cricket to travel in different directions, at different velocities while maintaining constant distance to a speaker.

Our spherical treadmill (Fig. 2) is comprised of four principal components: 1. a very-lightweight polystyrene ball that floats on a cushion of air, 2. a cowling that constrains the ball and maintains the air cushion around the ball, 3. an electronic sensor that detects rotation of the ball, and 4. software that records the sensor readings and calculates useful measures of the cricket's behaviour.

We chose the polystyrene ball based on its size (12.7 cm diameter) and small mass (11.7 g) which is c. 5× larger than a typical female cricket walking on its surface. The cowling (Figs. 2B and 2C) was designed using CAD software then rapid-prototyped in ABS

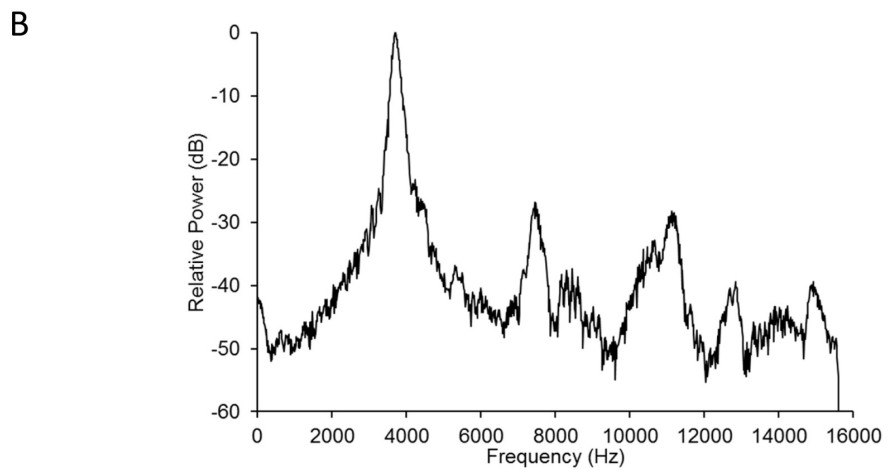

**Figure 1 Acoustic analysis of the standard mate attraction signal used for this study.** Oscillograms of a single pulse (top) and chirps (second from top) comprising the signal. A spectrogram (fourth from top) with time-aligned oscillogram of a complete chirp (third from top) shows the presence of slight harmonics in the signal. (B) FFT spectrum of a complete chirp showing the dominant frequency (highest peak) and harmonics (subordinate peaks to the right of the dominant). Harmonic content was c. 30 dB below the energy of the peak frequency. The signal was synthesized and stored as a 32 bit mono WAV file with sampling frequency of 31250 Hz. Spectrum was a 2048 point FFT with a Hanning window applied to the input signal. See text for additional details.

plastic. The dimensions of the cowling were chosen after the polystyrene ball was selected to ensure that the cowling had a small, c. 2 mm, gap between itself and the ball. Four holes were drilled through the cowling and outfitted with brass barbed hose fittings (Fig. 2D). Hoses attached to the fittings were then unified with a manifold and connected to an 11 L oil-less air compressor to ensure equal air pressure at symmetric locations beneath the ball. A collar was incorporated into the cowling design to prevent the ball from being ejected
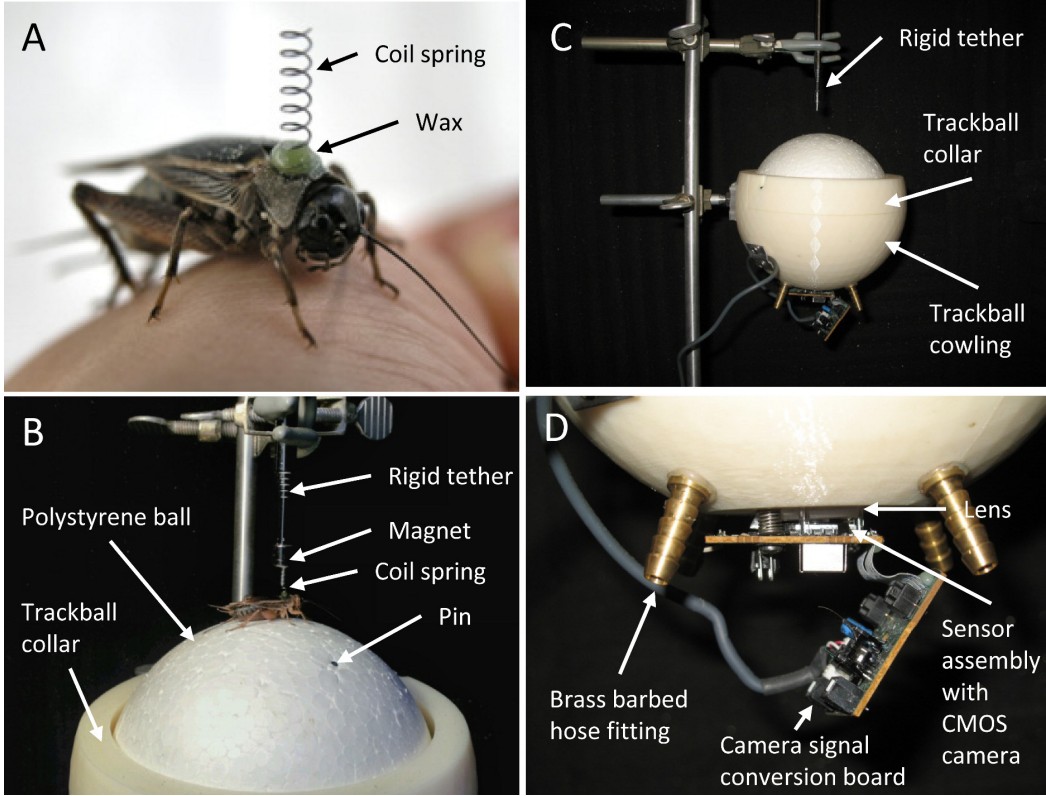

**Figure 2 Photographs of the mounting spring and spherical treadmill apparatus.** (A) A female cricket (*Gryllus assimilis*) with coil spring attached to her pronotum with wax. The spring is used to tether the cricket on the spherical treadmill while allowing her the freedom of body movements required for walking. (B) Female restrained on top of the spherical treadmill's polystyrene ball by her coil spring, the magnet and rigid tether. Note the metal dress-makers pin flush with the surface of the polystyrene ball; then pins were used to counter-act rotation motions caused by the ball's non-uniform density. (C) The polystyrene ball was suspended on a cushion of air and was constrained by the trackball cowling and collar. Air hoses are not attached in this photograph. (D) A magnified view of the electronic sensor assembly including the CMOS camera and lens used to detect rotation of the ball.

by the air. The air stream cushion beneath the ball effectively cancels the mass of the ball allowing the cricket to easily rotate the ball in multiple directions. The polystyrene ball is not uniform in density and had a tendency to 'roll' in one direction (without a cricket on its surface). To compensate for this, metal dress-makers pins were inserted into the ball until flush with the surface at appropriate locations to correct the centre of mass and thus counter-act these rotation motions (Fig. 2B).

To detect the motion of the ball, we extracted the sensor assembly, complete with USB cable, from a generic optical mouse. This sensor assembly is comprised of a pair of circuit boards with a complementary metal-oxide semiconductor (CMOS) camera and a clear plastic lens. We fixed the circuit board with the CMOS camera and lens to a window cut in the bottom of the cowling (Fig. 2D). The distance of the camera and lens from the surface of the polystyrene ball is critical for the proper function of the treadmill. We could adjust this distance with the screws used to attach the sensor assembly to the cowling.

We wrote custom software using MATLAB® (ver. 7.5, MathWorks, Natick, MA, USA) to detect the movement of the ball. Because the treadmill is essentially a functioning USB optical mouse, we simply used standard functions in MATLAB® for reading mouse pointer positions and sampled these data at a rate of 20 Hz during an experiment trial. The change in mouse position reported by MATLAB® is in pixel units. To convert between pixel units and mm, we simply applied a thin strip of label tape marked in mm increments to the ball surface and rotated the ball known distances in X and Y directions. This allowed us to determine a conversion factor from pixel units to mm of displacement. These control/calibration experiments also confirmed that the rotation of the ball produced linear increments in X and Y directions. Note, the calibration tape was removed for behavioural testing.

## Spherical treadmill trials

We ran all experimental trials in the dark under red light in a chamber (86 × 57 × 87 cm) lined with sound-attenuating foam. The spherical treadmill was placed in the middle of the chamber with two speakers 17 cm from the center of the sphere and 180° from each other. The speakers and the spherical treadmill were always set in the same location in the chamber.

We attached a coil- (micro-compression) spring (diameter: 3 mm, length: 8 mm; spring constant: 210.15 N/m) to the cricket's pronotum with low melting point wax on the day of the trial (Fig. 2A). Following a 10-min acclimatization period, we mounted the female on the spherical treadmill by attaching her spring to a magnetic rod (Fig. 2B). The spring and magnet facilitated rapid changing of specimens. Air pressure was adjusted such that the polystyrene ball, with the cricket in position, was able to rotate freely (without frictional contact with the cowling). Each female was able to freely rotate the polystyrene ball in all directions as she walked. All females were oriented in the same neutral position at the start of their trial: tethered on the spherical treadmill facing straight ahead between the two speakers. The magnet effectively kept the cricket oriented in the same direction. Following a 1 min silent acclimatization period, we presented the standard signal through a randomly selected speaker (to control for side bias). After 30 s of exposure to the standard signal, we recorded the female's phonotaxis relative to the broadcasting speaker. The standard signal was presented continuously throughout the 5 min trial, and the female's locomotor behavior was recorded from the treadmill at a sample rate of 20 samples per second. The 5 min trial therefore yielded a total of 6000 samples of cricket X, Y positions. We freed the female from the spring and returned her to the colony at the end of the trial. Temperature was held constant (between 22 and 23°C) in the trial room and was monitored with a Fisher Scientific Traceable Digital Thermometer. We also note that the compressor driving the spherical treadmill was located in a different room than the trials and the noise of the compressor was not audible to the crickets under test.

We calculated instantaneous displacement (cm) and velocity (cm/s) vectors from the positional data (X, Y coordinates). Total path length was calculated as the sum of all vector lengths over the 6000 samples. Net vector score (after *Huber et al., 1984*) was calculated

as the female's net movement toward or away from the standard signal during the 5 min trial. Net vector score takes into account the female's direction (vector angle) and the vector length of every recorded leg of movement:

$$\text{Net Vector Score} = \sum_{t=1}^{6000} [\cos(\text{vector angle } (t)) \times \text{vector length } (t)].$$

We defined the angle of the active speaker as 0°. Females moving directly toward the speaker (0°) had positive vector scores [$\cos(0°) = 1$], females moving directly away from the speaker (180°) had negative vector scores [$\cos(180°) = -1$], and females moving perpendicular to the speaker (90° or 270°) had vector scores of 0 [$\cos(90°$ or $270°) = 0$]. By multiplying this value by each vector length, and summing over the trial duration, we quantified the female's relative attraction to the standard signal. A large positive score indicated that the female moved quickly toward the speaker, a small positive score indicated the female moved slowly toward the speaker, and a large negative score indicated the female moved quickly away from the speaker (*Huber et al., 1984*).

### Statistical analysis

All data were analyzed using JMP® statistical software (ver 10.0.0, SAS Institute Inc., Cary, North Carolina, USA). Total path length and average instantaneous velocity magnitude were Box-Cox transformed because they were not normally distributed. We used one-way ANOVAs to quantify whether age influenced female response to acoustic stimuli. We used Tukey HSD tests to ascertain significant differences among the age groups when ANOVA results were significant. We corrected for multiple tests ($N = 4$) using *Benjamini & Yekutieli*'s (*2001*) false discovery rate ($FDR_{B-Y}$) method; our $FDR_{B-Y}$ corrected alpha was $P < 0.024$.

### RESULTS

Female age significantly influenced phonotaxis, explaining up to 64% of the variation among females (Table 1). Females 10- to 13-days post imaginal eclosion travelled longer total distance, walked faster and had higher net vector scores than females at most other ages. Age significantly influenced total path length (Table 1) and post hoc comparisons revealed that 10-day females moved further on the spherical treadmill than all age groups except 13-day females (Fig. 3A). Age also influenced average instantaneous velocity magnitude (Table 1). Post hoc comparisons revealed that 1- to 16-day females moved significantly faster than 22- to 28-day females, with 10- and 13-day females having slightly higher velocities than other females (Fig. 3B). Further, age significantly influenced net vector scores (Fig. 3C; Table 1). Post hoc comparisons revealed that females 1-, 4-, and 7-days had significantly lower net vector scores than females 10-days or older. While the velocity of females less than 19-days post imaginal molt was relatively high (Fig. 3B), 1- to 7-day females did not generally orient toward the speaker (Fig. 4). 10- to 16-day females both oriented toward the speaker and maintained fairly high velocities. Older females (22- to 28-day) oriented toward the speaker but at low velocities (Fig. 4).

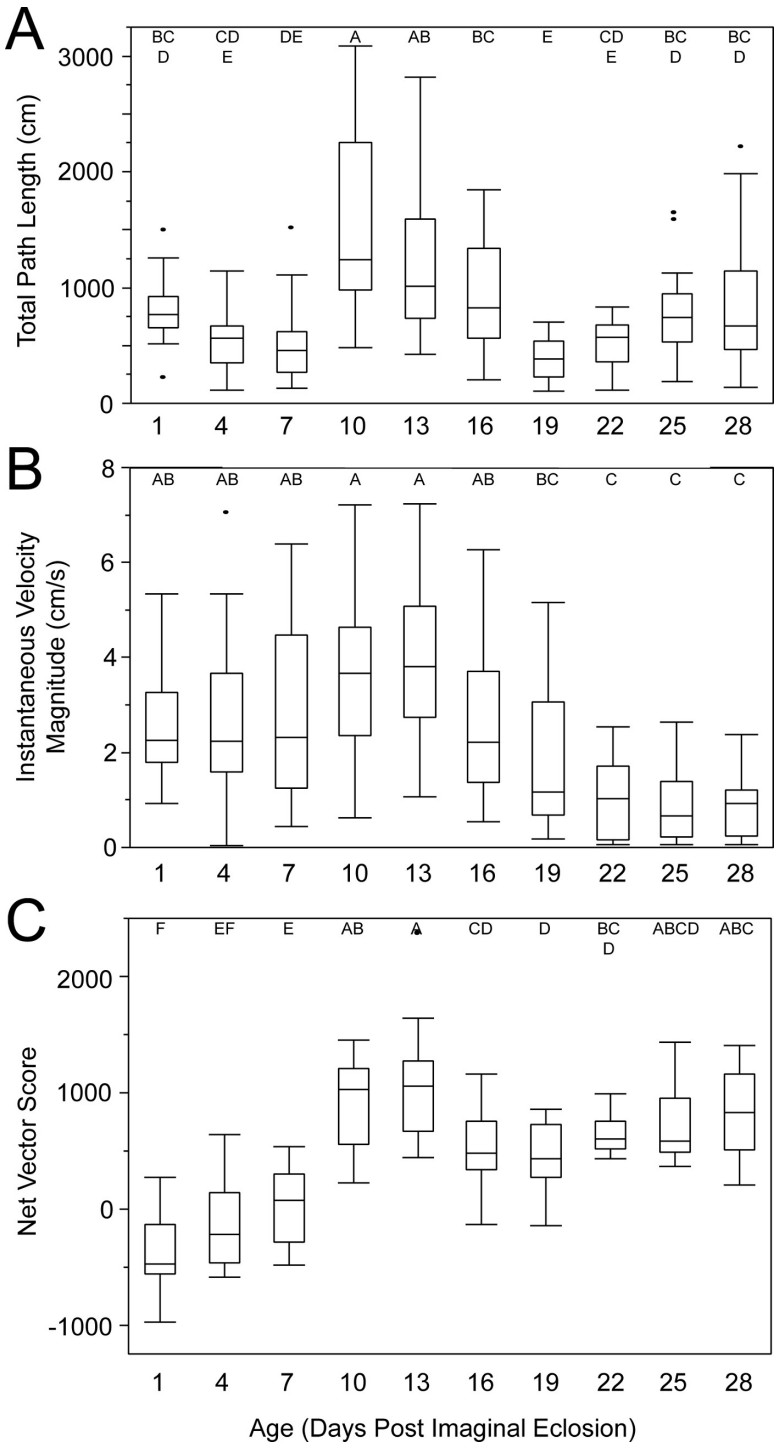

**Figure 3** **Box and whisker plots revealing how age influences female phonotaxis.** (A) total path length (cm), (B) instantaneous velocity magnitude (cm/s), (C) net vector scores. Letters above each age show significant differences (Tukey HSD post-hoc analysis).

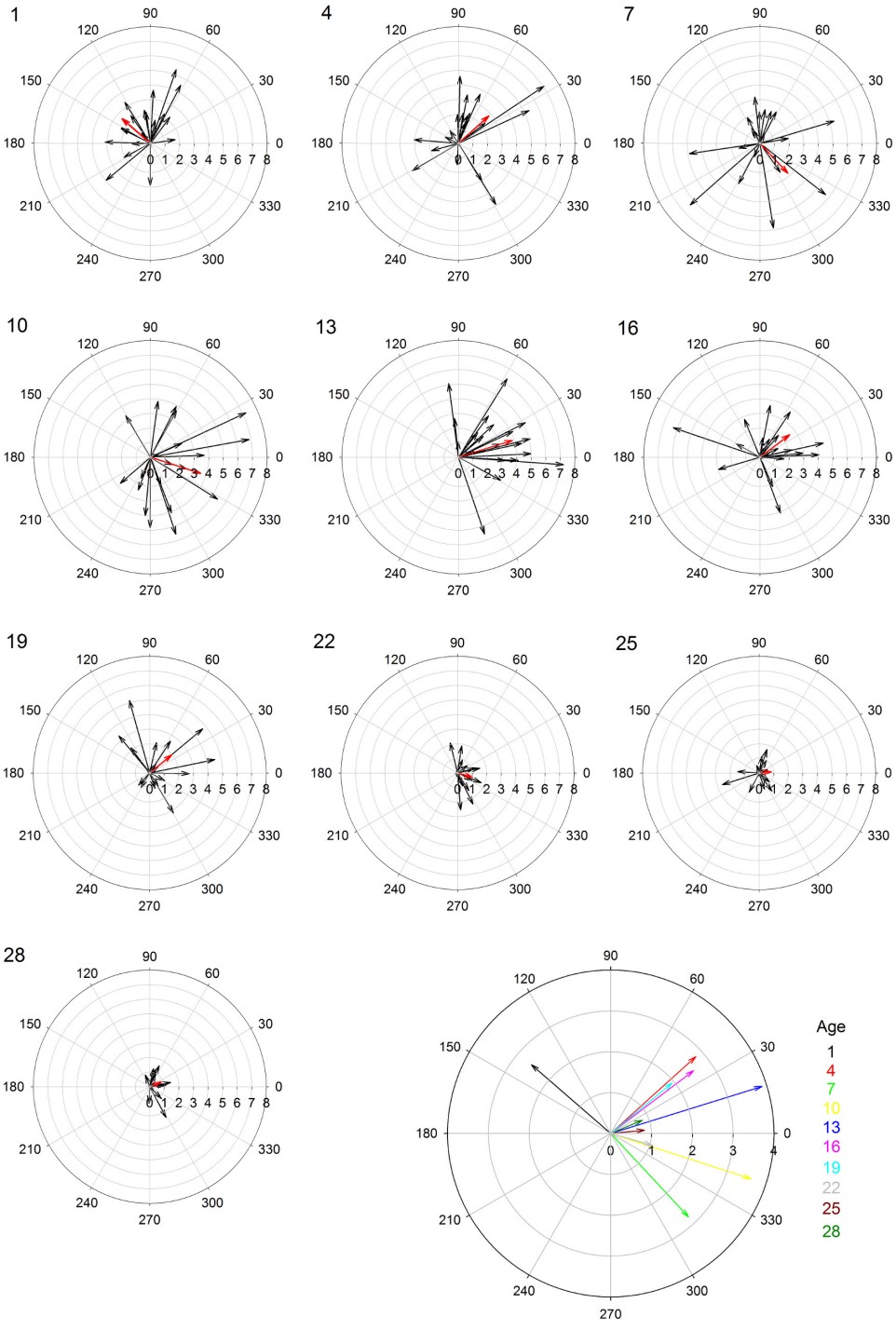

**Figure 4 Polar plots of female velocity (cm/s) when presented with the standard acoustic mate attraction signal.** The speaker broadcasted the standard signal from the right (0 degrees); we converted the vectors for trials where the standard signal was broadcasted from the left so they appear on the right. Inner circular grid-lines represent velocity magnitude (cm/s); tic marks around the outside of the 

**Figure 4 (...continued)**

circle indicate velocity angle; the number in the upper left hand corner indicates female age (days post imaginal eclosion). Red arrows show the mean velocity for all 20 females at that age group. The larger, multi-colored plot at the bottom shows the differences in average velocity and angle across all age groups.

**Table 1** ANOVA results showing the influence of age on phonotaxis measurements.

| Phonotaxis measurement | DF | *F* ratio | *P* value | $R^2_{adj}$ |
|---|---|---|---|---|
| Total path length (cm) | 1,9 | 13.2938 | <0.0001 | 0.3573 |
| Instantaneous velocity magnitude (cm/s) | 1,9 | 14.9834 | <0.0001 | 0.3874 |
| Net vector score | 1,9 | 40.2429 | <0.0001 | 0.6396 |

## DISCUSSION

In crickets, and other acoustic insects, phonotaxis plays an important role in mating, with females exhibiting positive phonotaxis toward the mate attraction signals of conspecific males (*Prosser, 1994*; *Cade, 1979*; *Sakaluk, 1982*; *Solymar & Cade, 1990*; *Jacot, Scheuber & Brinkhof, 2007*; *Jennions & Petrie, 1997*; *Ritchie, 1996*; *Verburgt, Ferreira & Ferguson, 2011*). We found age influenced female *G. assimilis* phonotaxis. Age explained 35–64% of the variation in phonotaxis behavior, with females 10–13 days post adult eclosion displaying the greatest movement toward the standard mate attraction signal compared to younger or older females. Our study suggests that when quantifying the preference landscape for *G. assimilis* sexual signals, 10- and 13-day females are likely to be the most responsive.

Some may criticise the use of a treadmill apparatus when quantifying female phonotaxis because the intensity of the stimulus does not change within a trial despite the locomotion behavior of the female. This is a valid criticism, as females are unable to triangulate and localize the sound source and may, therefore, not respond naturally. Regardless, we still found a statistically significant effect of age on orientation response and thus feel our results are a conservative estimate of age effects.

We had hypothesized that younger females would be more responsive than older females given their high residual reproductive value. While 1- to 7-day females traveled long distances and at high speeds, they did not orient toward the broadcast mate attraction signals (Fig. 4). Their high speed and distances traveled may signify higher energy reserves (*Prosser, 1994*), but their random orientation suggests little responsiveness towards the standard acoustic mate attraction signal. There are at least two possible explanations for their inattention to the signal: stronger preferences and/or lack of maturity. Young females may have not oriented to our standard signal because it was not attractive. Females often prefer signals with greater energetic content (*Wagner, Murray & Cade, 1995*) and extreme parameter values (*Ryan & Keddy-Hector, 1992*). Conversely, young females may have not oriented to our standard signal because they were not reproductively mature. Average age to initiate phonotaxis in *Acheta domesticus* is 5.4 days following imaginal molt, and females do not mate until they reached 6.9 days on average (*Prosser, 1994*). Other cricket studies also demonstrate onset of phonotaxis at 4–8 days post-imaginal molt

(*Walikonis et al., 1991*; *Loher et al., 1992*). Future research should present young females with multiple call stimuli to determine if they are reproductively immature or only responsive to signal parameters with extreme trait values.

10- and 13-day females exhibited greatest movement toward the standard signal. Older females also oriented toward the standard signal. These findings partially support our prediction that older females would show increased movement toward the standard signal compared to younger females. However, older females tended to travel at much slower speeds relative to younger females. Reductions in speed in older females may indicate senescence (*Prosser, Murray & Cade, 1997*). Overall, our findings are consistent with most studies on how age influences mating preference as we observed diminished responsiveness with increasing age (*Roff, 1992*; *Stearns, 1992*; *Gray, 1999*; *Olvido & Wagner, 2004*; *Mautz & Sakaluk, 2008*).

While our study suggests that age is an important indicator of phonotaxis, it is important to note that older females are unlikely to be virgin in the wild (*Sakaluk, 1982*; *Judge, Tran & Gwynne, 2010*; *Bretman & Tregenza, 2005*). *Judge, Tran & Gwynne (2010)* revealed that mating experience is more important than age in explaining female mating behavior. *Judge, Tran & Gwynne (2010)* used a fully factorial design where mated and unmated females of two different age classes were scored for their latency to mate with random males (no-choice mating paradigm). Given neither of our studies provided females with alternative acoustic stimuli, future research should use multiple acoustic stimuli to explore the relative importance of age versus mating experience on female preference functions.

## ACKNOWLEDGEMENTS

We would like to thank two anonymous reviewers for their comments on an earlier version. We would also like to thank Ryan Chlebak for designing and digitally rendering the spherical treadmill cowling and Andrew Mikhail who completed preliminary tests of the device. Finally, we would like to thank Sarah J. Harrison, Genevieve L. Ferguson, Ian R. Thomson, and Kathryn Dufour for their help with cricket care.

### Funding

Funding was provided by a Natural Science and Engineering Research Council of Canada Discovery Grants to SMB and JWD, the Canadian Foundation for Innovation Grants to SMB and JWD, the Ontario Research Fund to SMB and JWD, and Carleton University Research Fund to SMB and JWD. The funders had no role in study design, data collection and analysis, decision to publish, or preparation of the manuscript.

### Grant Disclosures

The following grant information was disclosed by the authors:
Natural Science and Engineering Research Council of Canada Discovery Grants.
Canadian Foundation for Innovation Grants.

Ontario Research Fund.

Carleton University Research Fund.

## Competing Interests

Susan M. Bertram is an Academic Editor for PeerJ.

## Author Contributions

- Karen Pacheco conceived and designed the experiments, performed the experiments, analyzed the data, wrote the paper.
- Jeff W. Dawson contributed reagents/materials/analysis tools, wrote the paper, wrote the Matlab code and co-designed the spherical treadmill.
- Michael Jutting co-designed and built the spherical treadmill.
- Susan M. Bertram conceived and designed the experiments, analyzed the data, contributed reagents/materials/analysis tools, wrote the paper.

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
