# Peer review of "How age influences phonotaxis in virgin female Jamaican field crickets (Gryllus assimilis)"

_PeerJ, doi:10.7717/peerj.130_

## Round 0.1 · original submission · Minor Revisions

Both reviewers have recommended minor revisions, and both reviewers have made some excellent points. I encourage the authors to carefully consider the reviewers' comments and to either implement or rebut those points. I look forward to the revised version of this manuscript and I would like to thank the authors for their submission to PeerJ, and the reviewers for their diligent efforts.

Reviewer 1 ·

Basic reporting

See general comments to authors.

Experimental design

See general comments to authors.

Validity of the findings

See general comments to the authors.

Additional comments

The manuscript by Pacheco et al. tests for an effect of female age on phonotaxis behaviour in the field cricket, Gryllus assimilis. They present a single male song stimulus to 200 different females of 10 different ages (N = 20 per age class) and measure 4 phonotactic behaviours using a non-compensating treadmill. They find that females at intermediate ages move faster and more directly toward male song than younger or older females.

Although I don’t have any major criticisms of the study (well-executed and well-written), I have to say that I was disappointed by the experimental design in that it seems fairly unambitious and a weak test of the hypothesis. The use of a single stimulus (with potentially the wrong parameters for the temperature broadcast – see specific comments below) will necessarily limit the generality and thus significance of the findings to the wider field. First off, a fully randomized design on its own is adequate, but not terribly powerful for testing age effects. A better design would have been to repeatedly test individual females to get a sense of whether individuals themselves change their preferences. As it stands now, the authors cannot quite say whether their results are an age effect per-se, or whether there is a selection effect such that females who survive a long time tend to have different phonotaxis behaviours. This confound makes it impossible (at least to me) to be sure that the main interpretation is correct.

Age-effects on female cricket phonotaxis is fairly well-travelled ground. I appreciate the work involved here and that this is a preliminary experiment to determine an optimal testing age for another study, but it would have been nice to see some more ambition (or to simply publish these results with the larger study).

Specific Comments

Line 15: The wording here seems awkward. Females “have” mating preferences, but they “make” mate choices. This is conventional usage in the literature.

Lines 42-43: What exactly do you mean by reproductive value here? Total fecundity? I think you mean “residual reproductive value” since it is really the fecundity that a female has remaining to her that declines with age. It is only because of the artificial lab situation of not allowing females to mate that total and residual fecundity are basically the same thing here (i.e. females haven’t laid any eggs yet and so they have yet to realize any fecundity). However, the original theory was surely not conceived to explain a pattern under extreme sperm limitation that you have here. Pianka and Parker (1975 Am Nat) would be an excellent paper to read on this theory.

Line 45: What exactly are the increased costs of being choosy that come with increasing age? It’s worth being explicit about these here as it will help the non-specialist reader.

Lines 90-91: Actually, using females only once is a very weak design for quantifying age effects because variation in phonotaxis behaviour cannot be partitioned into variation due to individual identity as you could if you used a repeated measures design. Consequently, as you see from your results, you can’t tell whether the variation you see in younger and very old females is due to those individuals having different individual preferences or to age causing the increased variation. Since you detected an age effect there is no real problem, but the study could have been much more powerful and yielded much more detailed results if it had been designed as a repeated measures (i.e. there were relatively large errors around response variables in each age class making trends hard to detect).

Lines 97-99: What life history models are you referring to?

Lines 100-102: I’m afraid I completely disagree with this statement. If I were planning a study to investigate the landscape of female preferences, I would not choose the age at which females showed the greatest consistency in phonotaxis. Wouldn’t you want evidence of variation? Also, the use of a single male song makes it difficult to generalize the results of this study.

Line 110: How was the water delivered?

Line 119: A volume measurement alone is not that informative – what were the dimensions?

Line 123: What was your rationale for choosing the ages that you did? Fewer age categories and more individuals per category would have given you greater power to detect age-based changes.

Lines 129-138: Could an interested reader replicate the signal from these parameters alone? What did the pulse sound envelope look like?

Line 137: SPL is a relative measure. What was your reference? I assume the standard sea-level sound pressure level standard, but it is good practice to put it in.

Lines 148-150: Why is maintaining a constant distance to a sound source a benefit? This method seems to introduce an high level of artificiality to the female cricket’s responses since she can’t actually localize the sound source and none of her actions result in the usual ability to triangulate. The chief benefit of using a Kramer treadmill seems to be to automatically yield numbers that can be used to quantify female phonotaxis (i.e. researchers don’t actually have to watch the females’ behaviours), but the interpretation of these numbers is not always clear. For instance, how do you distinguish search behaviour (i.e. zig-zagging to triangulate) from preference strength (see Wagner 1998 Anim Behav for an excellent discussion of this problem with mate choice studies). The treadmill apparatus is a very complicated device for such a simple question (as you can see, I’m not a fan – that said, the thing is pretty cool :)

Lines 156-157: Presumably the motivation was to approximate a flat surface without having so large a ball that it became impossible for the cricket to move. This raises the point of sphere weight – how heavy was your sphere? Others who have built these treadmills have worried about making the sphere light – how do you know that the sphere wasn’t too heavy to allow normal walking? Do you even need to worry about this since all of your crickets were treated the same way.

Line 172: What does CMOS stand for?

Lines 185-186: Excellent, very elegant.

Lines 197-199: Was the goal to allow females to rotate freely on top of the sphere without it recording her turning motions (i.e. the female could turn in place and the treadmill would not record this information if the ball didn’t move)? So this treadmill is different from some others that fix the animal pointing in one direction and then record turning motions. Sorry to be pedantic, but it is important to understand exactly what is being recorded so that the results can be interpreted and distinguished from other types of treadmill studies.

Line 211: The song parameters that you used to construct the standard song were averages of songs recorded at 26C (Whattam and Bertram 2011 Anim Behav), but your females were 3-4C cooler than the males that these songs represented. It is well-known that cricket song parameters change with temperature (Walker and Cade 2003 Can J Zool) so your song may not have represented the average male song at the temperature at which females were listening. How can you be sure that this mismatch between song and female temperature didn’t affect your results?

Line 227: The wording is a bit confusing here: “attractiveness” usually refers to the male song. Maybe this should be changed to “attraction”?

Lines 248-250: Do you mean inter-female variation, or did individuals vary within the trial? I assume you mean the former, but this should be made clear as your description makes it seem like individual females wandered randomly (i.e. unless you present some intra-individual measures of variability you cannot comment on individual consistency here).

Lines 252-254: I realize that your p-values are quite low, but you should make mention of repeated testing and institute some method of correcting for Type I error inflation.

Lines 260-261: I disagree with the interpretation that net vector scores increased “progressively” with age. No linear trend with age was tested, so a progressive increase cannot be evaluated.

Lines 263-268: I’m not clear on what this paragraph adds since the test of the vector scores addresses whether females went toward or away from the speaker. There are no statistics presented here so there is no basis on which to evaluate the authors’ claims. Also, figure 3 seems to simply be a presentation of the raw data for two out of the four response variables (angle and path length). Does this figure really need to be included?

Lines 299-307: This is where the language and terminology around mate choice gets a bit confusing. What is the rationale for interpreting reduced speed toward the signal as reduced choosiness? One could easily interpret reduced speed as increased choosiness since females are in effect requiring more signal to move a consistent distance toward the song source. This would have the effect of ensuring females approach only the highest effort signalers, which is what the vast majority of cricket phonotaxis seems aimed at. Some authors have used the term “responsiveness” to describe this speed measurement of phonotaxis (see Hunt et al. 2005 Am Nat; and Brooks and Endler 2001 Evolution for the prior paper).

Lines 312-314: This final sentence seems pretty boilerplate, but if Judge 2010 has already shown that mating experience is more important than age, then why do you say that future research should investigate this? Are you just talking about future research on Gryllus assimilis, or are you trying to be more general? Can you come up with something a bit more novel to suggest here (i.e. how would you improve Judge’s study)?

Figure 1: I like the photographs, but they seemed to be dorso-ventrally flattened in my version. Please ensure that they are not distorted.

Figure 3: Please ensure that the scale of the contour lines on these polar plots are the same for all ages. The figure is very difficult to interpret as it is now. Also, I think the last multicoloured plot has somehow gotten mixed up. Shouldn’t each of these lines correspond to a red line in the corresponding plot for that age (see lines 5213-515)? If so, they don’t match up (e.g. the black age 10 arrow is not the same length or direction as the red arrow in the age 10 plot above).

Table 1: I’m not sure about the rationale for including this table given that these are summary statistics for the entire sample of females and as such have no bearing on the hypothesis being tested. If the goal is to show variation, surely the error bars and different vectors do that in a much more informative way?

Reviewer 2 ·

Basic reporting

This paper reports female phonotaxis score as a function of female age in a field cricket. As PeerJ reviews papers based solely on "scientific validity" I have no problems with this paper, however I think that the authors should make some edits to more carefully distinguish between female "preference/selectivity" and phonotaxis score. The phonotaxis net vector scores used in this paper represent directed movement toward a single standard stimulus = "motivation." As alternate stimuli were not presented it is not possible to test selectivity or preference with these data. Nonetheless, I think that the data do adequately demonstrate the age group of females most likely to cooperate with the kugel trackball system in further trials.

Experimental design

Good for test of motivation/cooperation with test apparatus; inadequate for tests of preference.

Validity of the findings

Valid for motivation.

---

## Round 0.2 · accepted · Accept

Thank you to the referees for their thoughtful reviews of this MS, and to the authors for their comprehensive reply. I now recommend that this MS be published in PeerJ. As usual, I would also recommend that the authors opt to publish the reviews and their response alongside the article, as they serve to add context to the final publication.